# Shedding light on the nature of the catalytically active species in photocatalytic reactions using Bi$_2$O$_3$ semiconductor

Paola Riente [1✉], Mauro Fianchini [2], Patricia Llanes[2], Miquel A. Pericàs [2,3] & Timothy Noël [1,4✉]

The importance of discovering the true catalytically active species involved in photocatalytic systems allows for a better and more general understanding of photocatalytic processes, which eventually may help to improve their efficiency. Bi$_2$O$_3$ has been used as a heterogeneous photocatalyst and is able to catalyze several synthetically important visible-light-driven organic transformations. However, insight into the operative catalyst involved in the photocatalytic process is hitherto missing. Herein, we show through a combination of theoretical and experimental studies that the perceived heterogeneous photocatalysis with Bi$_2$O$_3$ in the presence of alkyl bromides involves a homogeneous Bi$_n$Br$_m$ species, which is the true photocatalyst operative in the reaction. Hence, Bi$_2$O$_3$ can be regarded as a precatalyst which is slowly converted in an active homogeneous photocatalyst. This work can also be of importance to mechanistic studies involving other semiconductor-based photocatalytic processes.

[1] Department of Chemical Engineering and Chemistry, Micro Flow Chemistry and Synthetic Methodology, Eindhoven University of Technology, Eindhoven, The Netherlands. [2] Institute of Chemical Research of Catalonia (ICIQ), The Barcelona Institute of Science and Technology (BIST), E-43007 Tarragona, Spain. [3] Departament de Quimica Inorgànica i Orgànica, Universitat de Barcelona, 08028 Barcelona, Spain. [4] Present address: Flow Chemistry Group, van't Hoff Institute for Molecular Sciences (HIMS), University of Amsterdam (UvA), Science Park 904, 1098 XH Amsterdam, The Netherlands.
✉email: p.riente.paiva@tue.nl; t.noel@uva.nl

Making the most of renewable resources is the order of the day. In this context, the use of sunlight as a perennial energy source to drive chemical transformations is at the forefront of this crusade[1]. Photocatalysis has emerged as a benchmark tool that combines light and a (photo-)catalyst to carry out chemical transformations that are otherwise elusive using standard synthetic procedures.

In the last decade, we are appreciating a fast growth in the use of photocatalysis for the synthesis of chemical compounds[2,3]. Along with it, a large variety of photocatalysts have been utilized which can convert sunlight into chemical energy and transfer this energy to the reacting molecules, thus effectively enabling light-fueled transformations. Amongst these diverse sets of photocatalysts, heterogeneous photocatalysis plays an important role in boosting the efficiency of photocatalytic systems, allowing them to recuperate and reuse the catalyst easily[4,5]. However, in contrast to their homogeneous counterparts, the catalytic processes involving heterogeneous photocatalysts are more complex and far from being fully understood. For instance, in most cases, identifying the true catalytically active species remains unsolved, and this is a key step towards the development of a reproducible and efficient heterogeneous photocatalytic approach, able to be used as a synthetic tool[6].

Heterogeneous photocatalysts based on metal oxide semiconductors have demonstrated high efficiency in carrying out a great variety of organic transformations[7,8]. In this regard, bismuth(III)oxide is a cheap and narrow bandgap metal oxide semiconductor ($E_g = 2.1$ to $2.8\,eV$) that presents useful photocatalytic activity. Due to the high oxidation power of the valence band hole ($\sim +3.13\,V$ vs. NHE), its photocatalytic efficiency was shown in several applications ranging from energy storage and photodegradation of dyes to biomedical applications[9,10]. Despite $Bi_2O_3$ exhibiting high efficiency in promoting photooxidations, the inability of its conduction band electrons ($\sim +0.33\,V$ vs. NHE) to interact with organic molecules results in fast electron-hole pair recombination which hinders application in reduction processes. Nevertheless, several studies have been reported on the enhancement of its activity by doping or tuning its surface[11–14]. In the last 5 years, $Bi_2O_3$ also became popular as a photocatalyst to drive light-induced organic transformations. Its photocatalytic activity was explored for the formation of C–C[15,16], C–S bond formation[17], and atom transfer radical addition (ATRA)-type reactions[18,19]. The interest in this semiconductor stems from its low price, non-toxicity, high availability, solid nature, and visible light response. Moreover, in some cases, it can replace the use of metal complexes based on expensive and non-abundant Ru and Ir transition metal photocatalysts[20,21]. Pericàs and co-workers pioneered the application of $Bi_2O_3$ as a photocatalyst in the asymmetric α-alkylation of aldehydes under visible light irradiation. The combination of enamine organocatalysis and $Bi_2O_3$ photocatalysis enabled access to a large variety of chiral aldehydes under mild reaction conditions[15]. After this successful application of $Bi_2O_3$ in photocatalysis, its use was also expanded to other transformations[16–19], such as the classical Kharasch reaction[22,23].

Importantly, in both the α-alkylation of aldehydes and the Kharasch addition, a phase change can be observed in the reaction mixture while the reaction progresses (Fig. 1). The reaction mixture shifts from a suspension to a transparent yellowish solution. Depending on the reaction type or/and substrates involved in the process, this change was faster or slower, and in most cases, it was directly associated with a successful formation of the desired product. Since $Bi_2O_3$ is not soluble in organic solvents[24], this phase change in the reaction mixture suggests that a soluble bismuth species could be formed during the photocatalytic process, which is most likely the actual photocatalyst. Although the exact structure of the soluble bismuth species might depend on the nature of the light-mediated process, we present here experimental and theoretical results, which provide key insights into the chemical nature of the catalytic species involved in the $Bi_2O_3$ photocatalytic processes. While we focused on bismuth-photocatalyzed processes, the results presented herein might have implications for other semiconductor-based photocatalytic reactions.

## Results and discussion

**Initial experimental observations**. To study the nature of the in situ-formed bismuth-species from $Bi_2O_3$, we selected the ATRA reaction between diethyl bromomalonate (DEBM) and 5-hexen-1-ol as a reaction model (Fig. 2a). As already mentioned, $Bi_2O_3$ was successfully used as a photocatalyst for this classical reaction under very mild reaction conditions, leading to the ATRA adducts in good to excellent yields in a process generally accompanied by the solubilization of the bismuth species[18]. A preliminary experiment was carried out to identify whether the substrate, the solvent, or the presence of light are somehow interacting with $Bi_2O_3$ triggering the formation of this soluble species (Fig. 2b). For that purpose, $Bi_2O_3$ was stirred in separate experiments with DEBM, 5-hexen-1-ol, and dry dimethyl sulfoxide (DMSO) under irradiation (white LED, ~ 23 W). The formation of a yellowish transparent homogeneous solution was observed when $Bi_2O_3$ was stirred together with DEBM and DMSO under irradiation for 10 h. To evaluate the optical absorption properties of the reaction mixtures, their UV–vis absorption spectra were recorded (Fig. 2c). Interestingly, the homogeneous mixture presented two new absorption bands in the near-ultraviolet region of the spectrum, a shoulder at $\lambda_{max} = 316$ and an intense band at $\lambda_{max} = 350\,nm$. We could independently establish that these bands do not correspond to any organic material present in the reaction medium or the combination of them under visible light irradiation (Supplementary Figs. 2 and 3). No changes in the reaction mixture took place unless DEBM and $Bi_2O_3$ were simultaneously present. In addition, the presence of light appears to be highly relevant for the appearance of the band at 350 nm (compare blue and green traces in Fig. 2c). To confirm the exact role of the soluble species arising from bismuth oxide in the photocatalytic ATRA reaction, DEBM and $Bi_2O_3$ were first stirred in dry DMSO overnight and subjected to irradiation. After the formation of the homogeneous solution, 5-hexen-1-ol was added and the reaction mixture was stirred an additional 24 h (Fig. 2d). Interestingly, full conversion was achieved, and not unexpectedly, no reaction took place in the absence of light (even after 8 days). Notwithstanding, after 8 days in the dark, the same reaction vessel was irradiated (24 h) and reached 68% of conversion into the ATRA product.

At this point, we can speculate that (i) $Bi_2O_3$ is not the photocatalyst for the ATRA reaction but is acting as a pre-catalyst; (ii) a soluble Bi-based species should be formed during the reaction course and; (iii) the as-formed soluble Bi-species is the active photocatalyst of the reaction.

**Importance of the nature of the substrate**. The appearance of new absorption bands in the UV–vis and the phase change of the reaction media were also observed in ATRA reactions carried out with other alkyl halide derivatives (Fig. 3). $Bi_2O_3$-promoted ATRA reaction performed with diethyl 2-bromo-2-methylmalonate (**II**), ethyl bromodifluoroacetate (**III**), ethyl bromoacetate (**IV**) and, ethyl bromofluoroacetate (**V**), under visible light irradiation, shifted to a yellowish homogeneous solution (ca. 10 h). For all these alkyl halides, absorption bands with very similar patterns and different intensities were detected near the visible region by UV–vis spectroscopy (Fig. 3a). In most

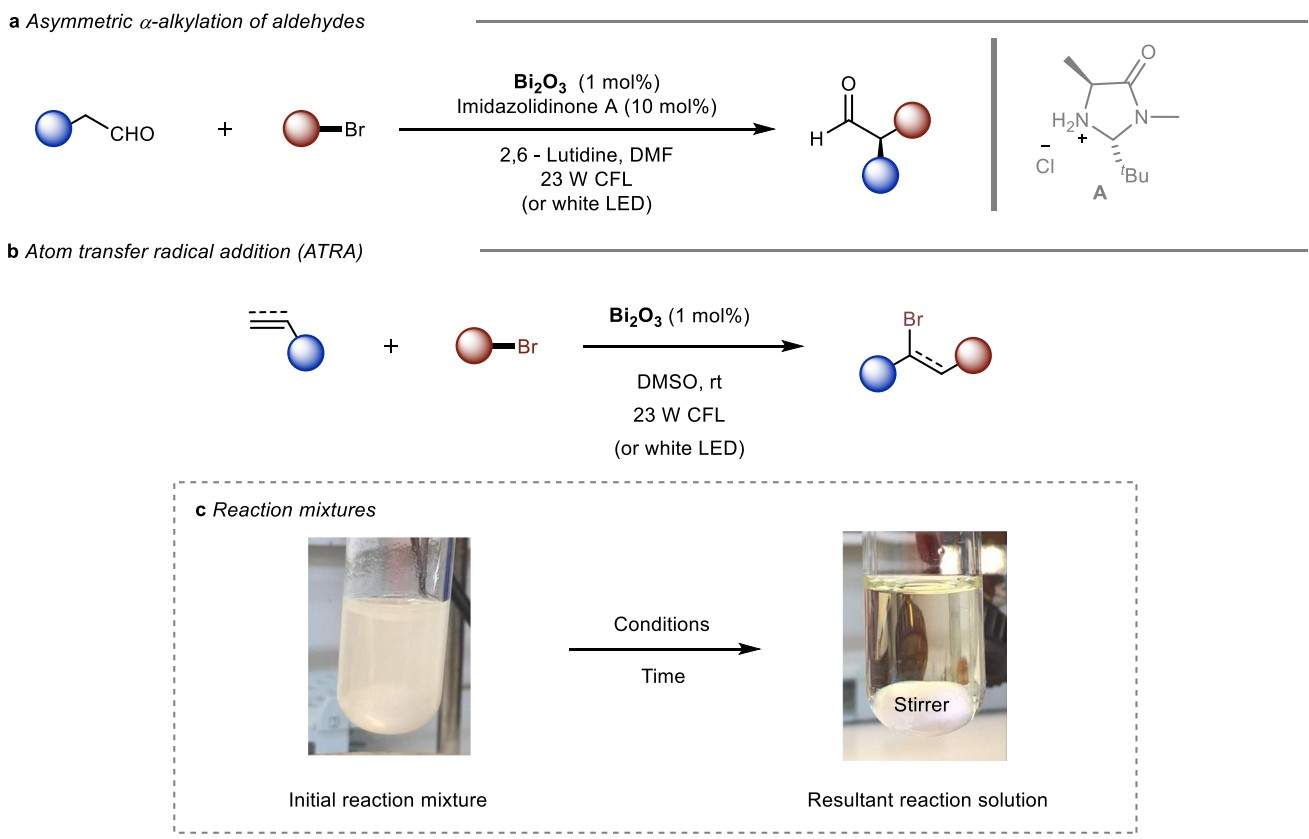

**Fig. 1 Unusual behavior of the Bi$_2$O$_3$ in photocatalysis. a** α-Alkylation of aldehydes photocatalyzed by Bi$_2$O$_3$. **b** Atom transfer radical addition (ATRA) reaction photocatalyzed by Bi$_2$O$_3$. **c** Representative pictures of the reaction mixtures for both reactions. CFL compact luminescent lamp, LED light-emitting diode.

cases, the ATRA reactions took place and reached full conversion after 24–48 h. However, no ATRA product was detected when the reaction was carried out with ethyl bromoacetate as a reaction partner. Also, no phase change occurred and this observation was also true with either diethyl chloromalonate (**VIII**) or diethyl fluoromalonate (**IX**). This result is in agreement with the fact that the reactivity of the alkyl halides is significantly affected by their bond dissociation and LUMO energies, alkyl bromides being better ATRA donors than alkyl chlorides/fluorides because C–Br bonds are more prone to homolytic cleavage than C–Cl and C–F ones[25,26]. Moreover, the observed affinity of solid Bi$_2$O$_3$ for secondary or tertiary alkyl bromides such as **I–III** and **V**, strongly suggests an interaction of these compounds with the Lewis acid sites present on the surface of the semiconductor[27], which triggers the homogenization process[28–30]. The simplest form of the complex formed during the reaction might be a coordination of the bismuth (III) to the carbonyl donor group. However, the soft Lewis acid character and the halogenophilic properties of bismuth may result in the coordination with the bromine atom[31,32]. Based on these reports, direct interaction with the bromine atom could also be envisaged. For that purpose, the ATRA reaction was carried out using 2-bromooctane (**VI**) and CBr$_4$ (**VII**) as non-carbonyl compounds. As expected, both the conversion and the phase change were not observed for the ATRA reaction with 2-bromooctane after 72 h of reaction time (the standard dissociation energy (DH$_{298}$) for CH$_3$–Br is *ca.* 70 kcal/mol)[33]. The UV–vis analysis of the reaction mixture revealed two bands with very low intensity. In contrast, full conversion was observed with CBr$_4$ together with the appearance of the corresponding absorption bands with significant intensity in the UV–vis spectra

(DH$_{298}$ for CBr$_3$–Br is *ca.* 50 kcal/mol). Further, when the reaction was performed with a non-brominated ester, such as diethyl malonate (**X**), both absorption bands and the phase change were not observed (for this substrate the reaction was carried out in the absence of the olefin). These results indicate that the formation of the Bi-soluble species is associated with the coordination of bismuth to bromine atoms, rather than to the carbonyl oxygen moiety. This triggers the solubilization process and the appearance of the new absorption bands in the UV–vis spectra. A similar interaction with α-bromo esters was also observed in the presence of other solvents (Supplementary Fig. 4).

**Kinetic experiments and UV–vis studies.** We next followed over time the formation of soluble intermediate bismuth species and the targeted ATRA adduct in the reaction of 5-hexen-1-ol with donor **I** (Fig. 4a) by UV–vis spectroscopy and $^1$H nuclear magnetic resonance (NMR), respectively. To this end, aliquots of the reaction mixture were taken every 2 h to check the reaction progress. The UV–vis spectra of the reaction mixture showed that the expected absorption bands gained in intensity over time (Fig. 4b). In a parallel manner, the analysis of the same sample by $^1$H NMR revealed the formation of the ATRA product as soon as the soluble Bi-species is formed. Moreover, for the kinetic profile of the reaction, an S-shaped curve was obtained caused by a long induction period, suggesting that the formation of an intermediate species is responsible for the light-induced formation of the ATRA product (Fig. 4c). Similar behavior was observed when the reaction was carried out in presence of dimethylformamide (DMF) (Supplementary Fig. 4e). This data also provides pieces of evidence that the formation of the soluble species is crucial for the

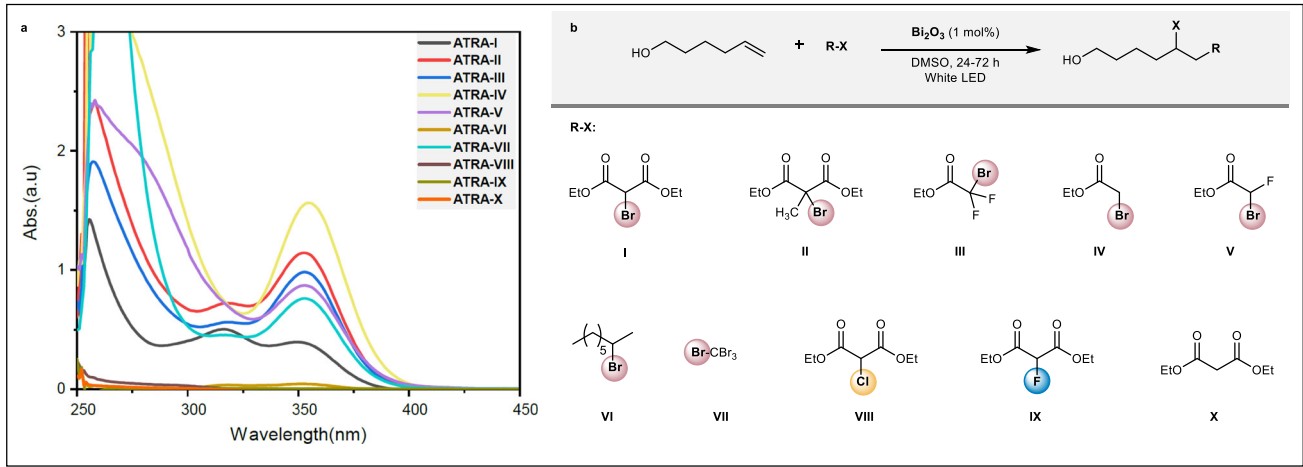

**Fig. 2 ATRA reaction photo-induced by Bi₂O₃. a** Reaction model. **b** Behavior of the Bi₂O₃ with substrates and solvent. **c** UV–vis absorption spectra of the reaction mixtures. The indices used in the plot are defined in (**a**, **b**). **d** Light, and dark experiments for the ATRA reaction carried out in two steps. Reaction conditions: Bi₂O₃ (1 mol%), DEBM (1 equiv.), 5-hexen-1-ol (1.1 equiv.), DMSO (2 mL).

**Fig. 3 Substrate probe for the Bi₂O₃-induced ATRA reaction. a** UV–vis analysis of the reaction crudes in the Bi₂O₃-promoted ATRA reaction of 5-hexen-1-ol with a variety of ATRA donors (**I–X**). **b** Employed ATRA donors. Compounds **VI**, **VIII**, **IX**, and **X** failed to react.

ATRA reaction using Bi₂O₃ since, in its absence, no formation of ATRA adducts was observed.

Another important aspect is the loss of heterogeneity of the reaction. Bi₂O₃ is an insoluble compound in water and organic solvents. Its solubility is observed only in harsh conditions, such

as in an excess of HBr. However, this oxide can be solubilized in acid chlorides through the cleavage of the Bi–O bonds leading to in situ generations of BiCl₃ and the corresponding acid anhydrides[34]. This behavior was also observed with other Bi-based oxides, such as BiOCl[35]. Based on these precedents, we

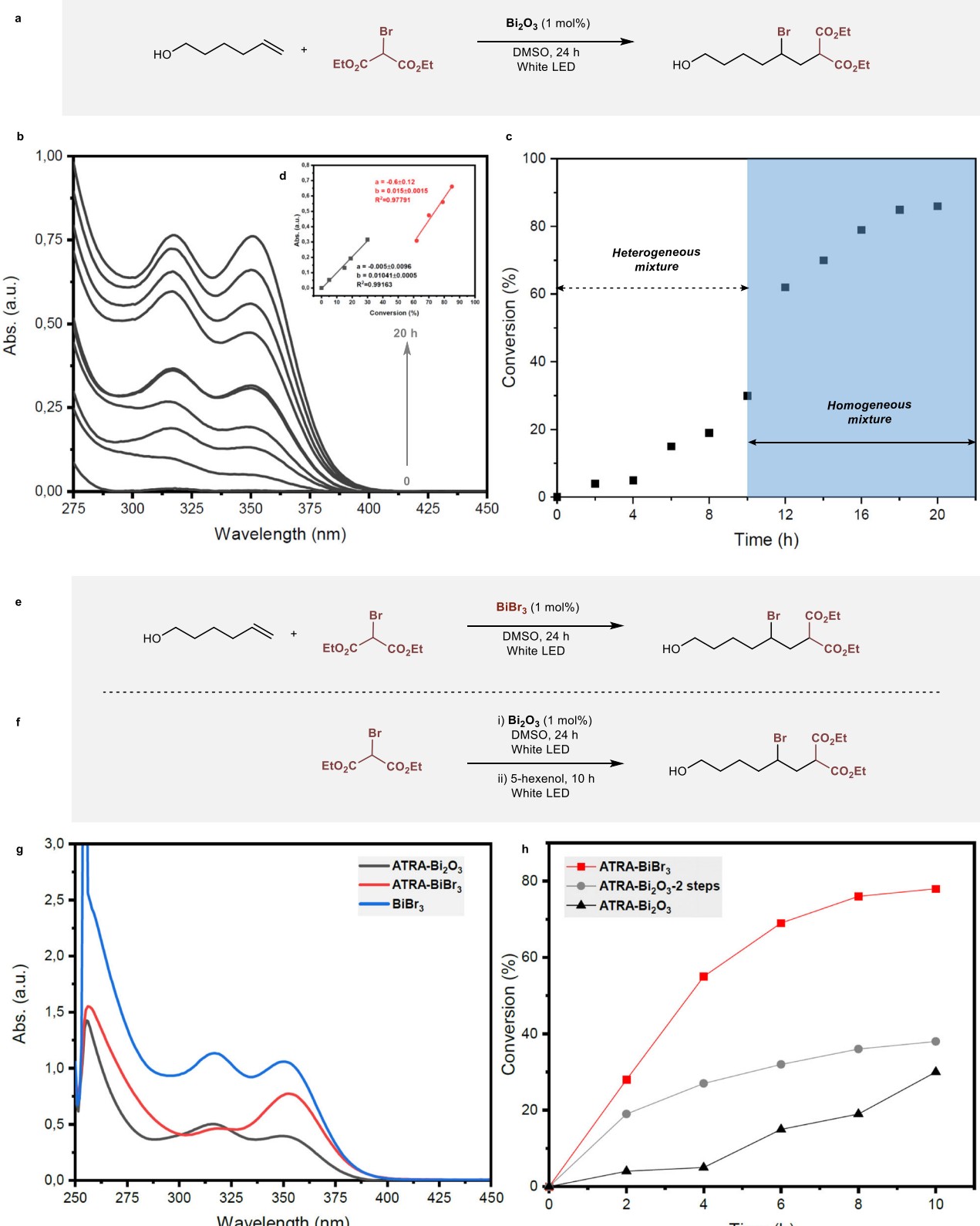

**Fig. 4 UV–vis spectroscopy and $^1$H NMR studies for the ATRA reaction. a** ATRA reaction induced by Bi$_2$O$_3$. **b** Spectral evolution of the absorption of a DMSO solution of Bi$_2$O$_3$ (5 mM) with 0.5 M diethyl bromomalonate and 0.55 M of 5-hexen-1-ol under irradiation. **c** Kinetic plots of conversion versus time for the ATRA reaction in the presence of Bi$_2$O$_3$. **d** Inset: Linear relation between absorbance versus conversion in the heterogeneous (black) and homogeneous regimes (red). **e** ATRA reaction induced by BiBr$_3$. **f** ATRA reaction carried out in two steps. **g** UV–vis spectra from the ATRA reaction in the presence of Bi$_2$O$_3$ and BiBr$_3$. **h** Comparison of the ATRA reaction profile for Bi$_2$O$_3$ and BiBr$_3$ under irradiation. Conversions were obtained by $^1$H NMR of the reaction mixture.

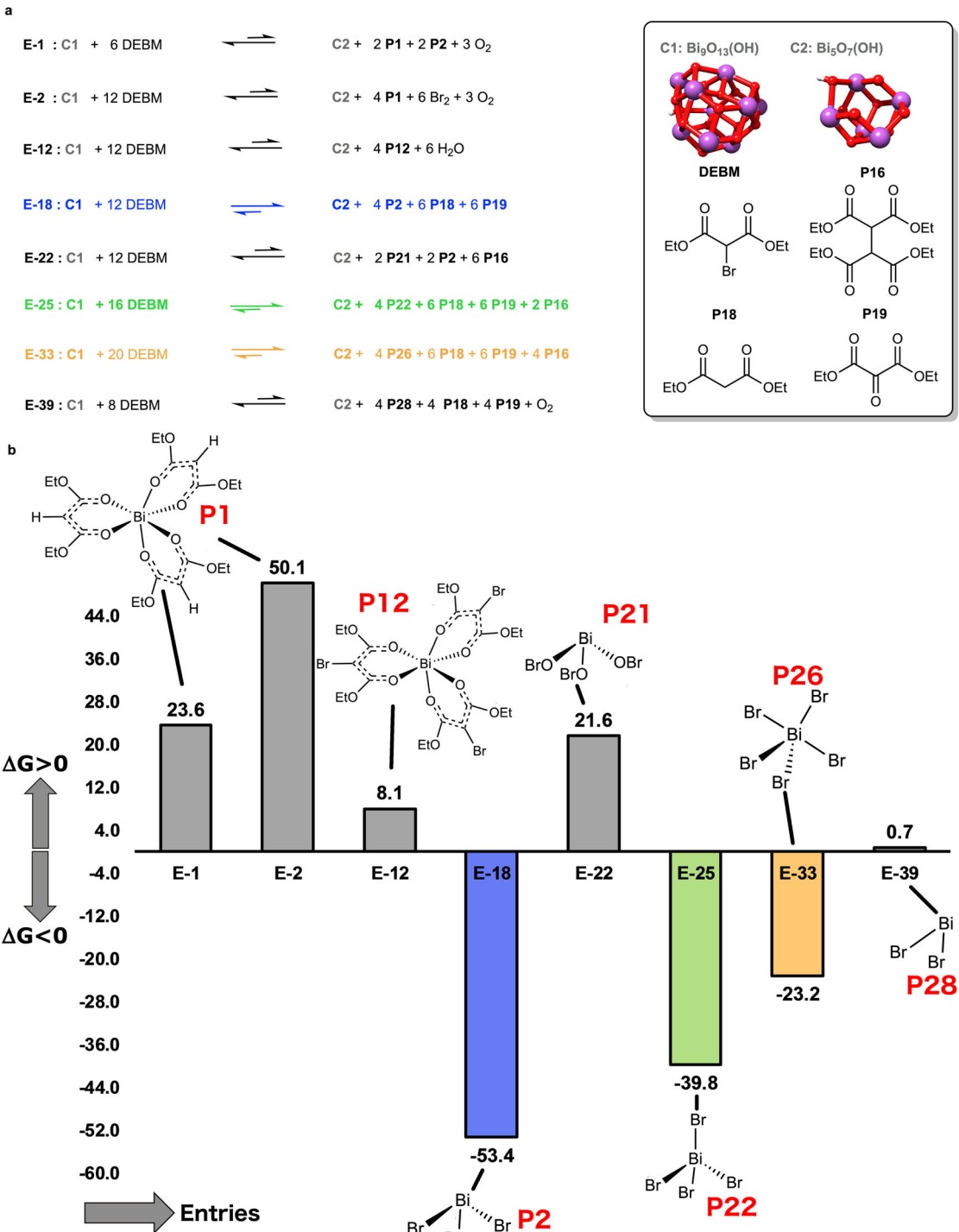

**Fig. 5 Mapping thermodynamic stability of soluble bismuth complexes. a** Selected equilibria for most relevant reactions from [$Bi_9O_{13}(OH)$] nanomer and DEBM included in Dataset 0-3. **b** Gibbs free energy of formation ($\Delta G^0$) for the selected products at point (**a**) in DMSO at 298 K (1 M standard state). The data are in kcal mol$^{-1}$ per equivalent of bismuth.

envisaged that a soluble species of $Bi^{3+}$ would be formed in the reaction medium through its interaction with labile Br–C bonds, leading to the appearance of the observed UV band. $Bi^{3+}$ is the most common and stable ionic form of bismuth and it is often used as an activator or sensitizer. Moreover, it always shows two absorption bands in the UV region, viz., the transitions $^1S_0$–$^3P_1$ and $^1S_0$–$^1P_1$[36,37]. To investigate the presence of a soluble species of $Bi^{3+}$, we decided to use $BiBr_3$ as a model in the ATRA reaction. $BiBr_3$ is a highly hygroscopic yellow salt and tends to form

adducts or $s^2$ coordination complexes with Lewis base ligands, this being the origin of its solubility in some polar aprotic organic solvents, such as DMF and DMSO[38]. The UV–vis spectrum of $BiBr_3$ in DMSO showed essentially identical absorption spectra observed for the $Bi_2O_3$-induced ATRA reaction, namely two $\lambda_{max}$ values around 312 and 350 nm (Fig. 4g). Also, to confirm our hypothesis, the ATRA reaction was carried out using $BiBr_3$ as a photocatalyst under the same standard conditions. The reaction was stirred overnight and the resulting solution was analyzed by

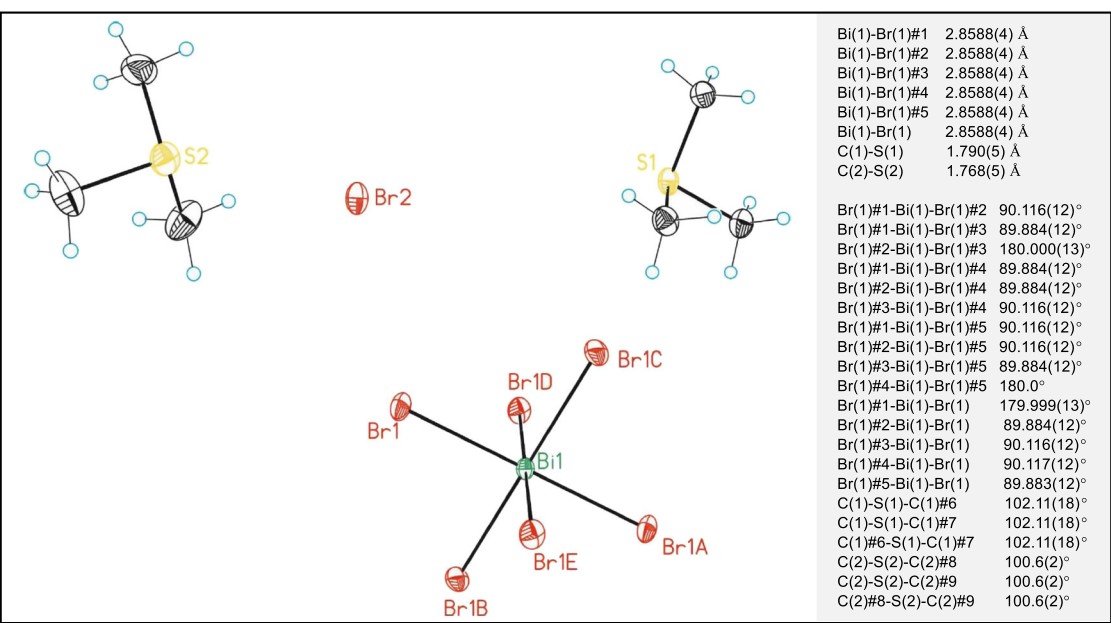

| | |
|---|---|
| Bi(1)-Br(1)#1 | 2.8588(4) Å |
| Bi(1)-Br(1)#2 | 2.8588(4) Å |
| Bi(1)-Br(1)#3 | 2.8588(4) Å |
| Bi(1)-Br(1)#4 | 2.8588(4) Å |
| Bi(1)-Br(1)#5 | 2.8588(4) Å |
| Bi(1)-Br(1) | 2.8588(4) Å |
| C(1)-S(1) | 1.790(5) Å |
| C(2)-S(2) | 1.768(5) Å |

| | |
|---|---|
| Br(1)#1-Bi(1)-Br(1)#2 | 90.116(12)° |
| Br(1)#1-Bi(1)-Br(1)#3 | 89.884(12)° |
| Br(1)#2-Bi(1)-Br(1)#3 | 180.000(13)° |
| Br(1)#1-Bi(1)-Br(1)#4 | 89.884(12)° |
| Br(1)#2-Bi(1)-Br(1)#4 | 89.884(12)° |
| Br(1)#3-Bi(1)-Br(1)#4 | 90.116(12)° |
| Br(1)#1-Bi(1)-Br(1)#5 | 90.116(12)° |
| Br(1)#2-Bi(1)-Br(1)#5 | 90.116(12)° |
| Br(1)#3-Bi(1)-Br(1)#5 | 89.884(12)° |
| Br(1)#4-Bi(1)-Br(1)#5 | 180.0° |
| Br(1)#1-Bi(1)-Br(1) | 179.999(13)° |
| Br(1)#2-Bi(1)-Br(1) | 89.884(12)° |
| Br(1)#3-Bi(1)-Br(1) | 90.116(12)° |
| Br(1)#4-Bi(1)-Br(1) | 90.117(12)° |
| Br(1)#5-Bi(1)-Br(1) | 89.883(12)° |
| C(1)-S(1)-C(1)#6 | 102.11(18)° |
| C(1)-S(1)-C(1)#7 | 102.11(18)° |
| C(1)#6-S(1)-C(1)#7 | 102.11(18)° |
| C(2)-S(2)-C(2)#8 | 100.6(2)° |
| C(2)-S(2)-C(2)#9 | 100.6(2)° |
| C(2)#8-S(2)-C(2)#9 | 100.6(2)° |

**Fig. 6 X-ray ORTEP structure of [BiBr$_6$][(CH$_3$)$_3$S]$_3$ with representative bond distances and angles.** Bismuth showed in green, bromide in red, sulfur in yellow, carbon in black, and hydrogen atoms in blue. This mixture crystallizes in a trigonal R$\bar{3}$c space group with cell parameters $a = b = 9.706$ Å, $c = 55.160$ Å, $\alpha = \beta = 90°$, and $\gamma = 120°$ (see Supplementary Section 4 for further details).

[1]H NMR and UV–vis spectroscopy (Fig. 4g, h). Gratifyingly, the ATRA reaction using BiBr$_3$ reached full conversion into the ATRA product in a shorter reaction time. Also, the UV–vis analysis of the reaction mixture exhibited two absorption bands with a similar pattern than those observed with Bi$_2$O$_3$-derived samples. Figure 4h shows the conversion into the ATRA adduct using BiBr$_3$ as the photocatalyst. Interestingly, no induction period was observed in that case. Finally, a third experiment was performed to compare the kinetics of the reaction after its shifting into a homogenous mixture using Bi$_2$O$_3$ subjected to irradiation. With this aim, the reaction was carried out in two stages as shown in Fig. 4f. Again, no induction period was observed and the reaction presented essentially the same kinetic profile as those observed in the presence of BiBr$_3$. These results strongly support that Bi$_2$O$_3$ acts as a pre-catalyst and its interaction with bromo-derivatives lead to the formation of a soluble Bi$_n$Br$_m$-species triggering the photocatalytic process[39].

**Computational studies on the formation of soluble Bi$^{m+}$ species.** To provide a rational justification to our claim on the formation of soluble species of BiBr$_3$, we performed theoretical calculations at M06-L-D3/def2-TZVPP//M06-L-D3/def2-SVP level in DMSO via the solvation model based on density scheme (see Supplementary Information, Section 5 for further details). Figure 5 summarizes the computed free energy ($\Delta G^0$) for eight equilibria involving [Bi$_9$O$_{13}$(OH)] nanomer (C1) and DEBM (see Supplementary Information for specific entries at pp. 17–24). In each equilibrium two Bi$_2$O$_3$ units from C1 react with DEBM via redox or acid–base pathways to form four soluble bismuth species, a residual [Bi$_5$O$_7$(OH)] cluster (C2), and different sub-products. The eight entries shown in Fig. 5 have been selected as the most representative ones from extended Datasets 0–3, comprehensive of 48 entries reported in the computational section of the Supplementary Information. E-1 and E-2 in Fig. 5 clearly show that both the partial (two equivalents, E-1) and the total (four equivalents, E-2) formation of a Bi$^{3+}$–O chelated complex (P1) via redox reactivity results to be thermodynamically unfeasible (for more entries see Supplementary Information, pp.

17–18). Entry E-12 clearly shows that even an acid-base pathway forming a Bi$^{3+}$–O chelated complex (P12) is thermodynamically unfeasible. Entry E-18 describes the complete formation of four equivalents of BiBr$_3$ (P2) with concomitant formation of diethyl malonate (P18) and diethyl ketomalonate (P19) as sub-products. Traces of these sub-products (P18 and P19) have been detected in the reaction medium by gas chromatography–mass spectrometry (see Supplementary Fig. 6). Entry E-18, with $\Delta G^0 = -53.4$ kcal mol$^{-1}$ per equivalent of bismuth, is the most right-shifted equilibrium (i.e., toward the formation of the products) that we encountered in the whole investigated data pool. Besides, it is worth mentioning that the formation of four equivalents of BiBr$_3$ results to be always thermodynamically feasible, independently of the nature of the generated malonate sub-products (e.g. tetraethyl 1,1,2,2-ethanetetracarboxylate or tetraethyl 1,1,2,2-ethenetetracarboxylate, see Supplementary Information at pp. 19). Entries E-25, E-33, and E-39 describe the equilibria for the formation of BiBr$_4$ (P22), BiBr$_5$ (P26), and BiBr$_2$ (P28), respectively; even though the formation of bismuth bromides, with Bi in a different oxidation state than III, is generally thermodynamically possible, we see from Fig. 5b that it is never more favored than Bi$^{(III)}$Br$_3$ (see also Supplementary Information at pp. 19–20). Hence, since the formation of homogeneous BiBr$_3$ is theoretically suggested, we turned our attention to the crystallization of an irradiated solution of Bi$_2$O$_3$, DEBM, and DMSO, to gain deeper insights into the photocatalytic system. The obtained crystals were analyzed by single-crystal X-ray diffraction revealing an interesting mixture composed of [(BiBr$_6$)]$^{n-}$ octahedral anions balanced by [(CH$_3$)$_3$S]$^+$ cations and [(CH$_3$)$_3$S]Br (Fig. 6). The presence of trimethyl sulfonium bromide is in good agreement with the room-temperature decomposition of DMSO in the presence of α-halo carbonyl or ester groups reported by Kornblum[40], Hess[41], and co-workers. Similar species were confirmed by UV–vis spectroscopy in a reaction mixture containing BiBr$_3$, DEBM, and trimethyl sulfonium bromide (Supplementary Fig. 5).

Our calculations support the thermodynamically feasible formation of solvato-complexes from BiBr$_3$, showing DMSO in the primary sphere of coordination of bismuth (Fig. 7 entries

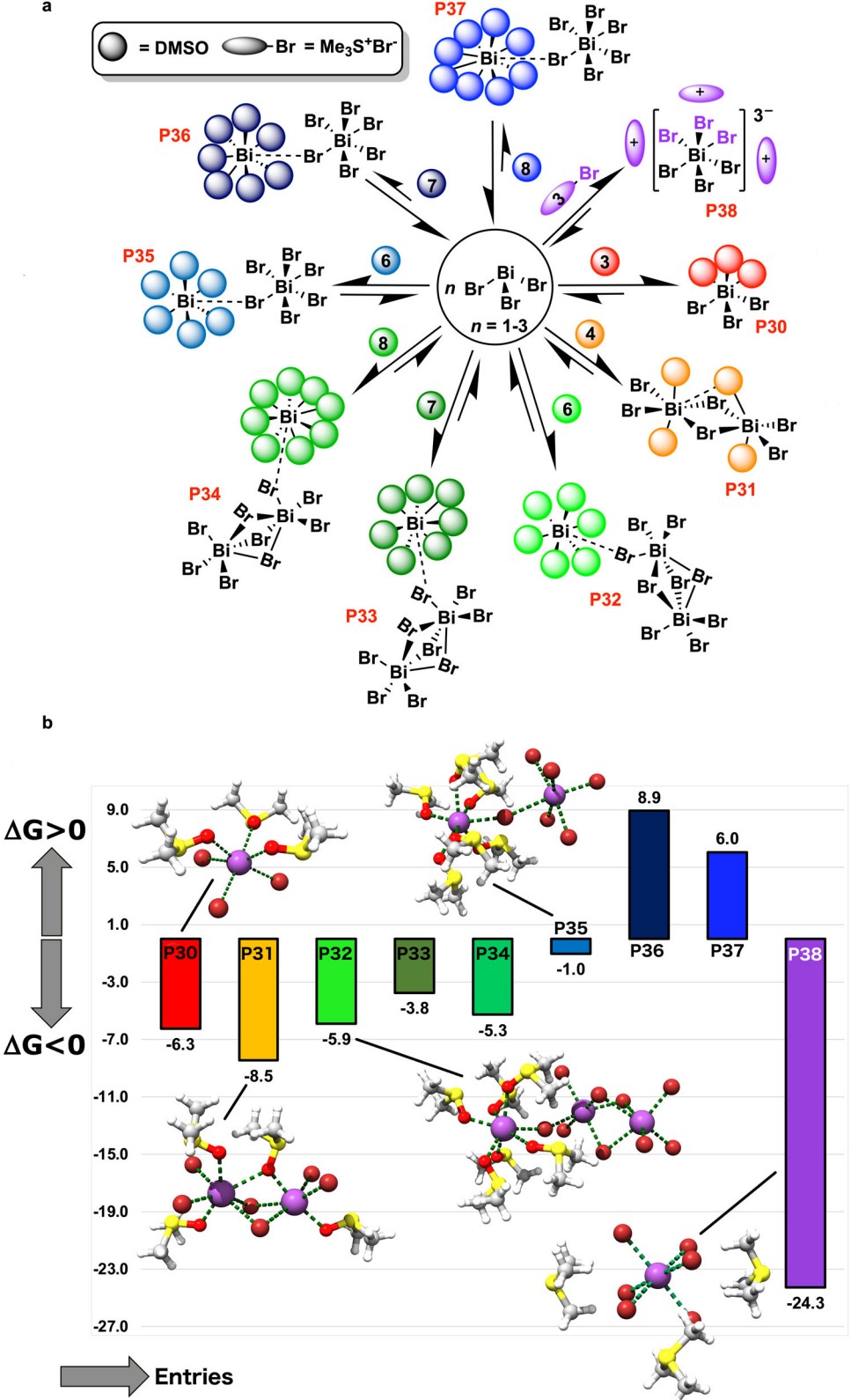

**Fig. 7 Improving the realism of the model: solvation of BiBr₃ versus complexation with (Me₃S)Br. a** General reaction scheme for solvation of naked BiBr₃ with dimethyl sulfoxide (sketched as a sphere for clarity) or complexation with (Me₃S)Br (sketched as an ellipsoid for clarity), the salt formed via Kornblum–Hess degradation of DMSO in presence of DEBM. **b** Free energy of formation (ΔG⁰) of DMSO-adducts (P30–P37) and {[(Me₃S)₃][BiBr₆]} adduct (P38) at 298 K and 1 M standard state. The data are in kcal mol⁻¹ per equivalent of bismuth. Relevant molecular structures are overlaid within the graph (Bi in purple, Br in dark red, O in bright red, S in yellow, C in gray, and H in white).

P30–P37). Similar structures (i.e., bearing different halogens) have been previously isolated and characterized via X-ray crystallography[42] (calculated metric parameters are in good agreement with experimental counterparts). It is logical to postulate that solvato-complexes of similar structures can be formed for $BiBr_3$ in presence of other strong σ-donor solvents like acetonitrile or DMF (e.g., DMF provides an additional calculated stabilization to $BiBr_3$ of $-3.9$ kcal mol$^{-1}$ in fac-$(DMF)_3BiBr_3$, while DMSO provides additional $-6.3$ kcal mol$^{-1}$ in fac-$(DMSO)_3BiBr_3$ (entry P30 in Fig. 7), mirroring the different affinities of the solvents towards bismuth). Moreover, calculations elegantly explain why we were not able to isolate solvato-complexes of $BiBr_3$ in DMSO in presence of DEBM. $[(CH_3)_3S]Br$ salt, coming from the degradation of DMSO, acts as a ligand, displacing ligated molecules of DMSO with Br$^-$ anions (for their high affinity toward Bi$^{3+}$) and leads to the formation of the thermodynamic product $[(CH_3)_3S]_3[(BiBr_6)]$ (Fig. 7, entry P38), that is, in fact, ~16 kcal mol$^{-1}$ more stable than the most stable solvato-complex (Fig. 7, entry P31). Though we could not estimate the oscillator strengths of spin-forbidden transitions to get more accurate theoretical versus experimental assignments, calculated singlet-to-triplet transitions correlate qualitatively well with the experimentally obtained UV–vis spectra (see Supplementary Information at pp. 45–46). For more details on the atomic cartesian coordinates, harmonic frequencies, RRHO-corrected, and non-corrected energies for all the stationary points reported in the present work see Supplementary Data 1.

**Conclusion**. In summary, we have elucidated the catalytically active species involved in photocatalytic processes where $Bi_2O_3$ is used. Our combined theoretical and experimental studies revealed that the most stable species, formed from a reaction between $Bi_2O_3$ and certain alkyl bromides, are closely related to pure $BiBr_3$ or $BiBr_3$- based complexes in the presence of DMSO or DMF. These species can absorb light, which triggers the subsequent formation of the required alkyl radical in ATRA and α-alkylation reactions. Even though $Bi_2O_3$ serves as a precatalyst, $Bi_2O_3$ is a non-hygroscopic, cheap, and innocuous compound. Therefore, it offers numerous advantages over the direct use of the highly hygroscopic $BiBr_3$. The homogenization of such easy-to-handle heterogeneous $Bi_2O_3$ catalyst may have its implications to other photocatalytic systems and is of importance for transferring this process to continuous-flow where the handling of suspensions remains challenging due to clogging phenomena[43–45]. We anticipate that this work could stimulate other mechanistic research when using metal oxide semiconductors as photocatalysts.

## Methods

**General procedure for the ATRA reaction**. To a sealed vial filled with argon, containing $Bi_2O_3$ powder (Sigma-Aldrich, powder, 99.999% traces metal base, 4.7 mg, 0.01 mmol) or $BiBr_3$, (Sigma-Aldrich, anhydrous powder, 99.998% trace metals basis, 4.5 mg, 0.01 mmol), the corresponding organobromine (1.0 mmol) and degassed dry solvent (2 mL) was added through a septum. To this suspension, the alkene (1.1 mmol) was added via syringe, and the mixture was degassed for 10 min by bubbling argon through the reaction medium (for the volatile substrate the reaction vessel was poured into an ice bath). Inlet and outlet needles were removed; the vial was sealed (parafilm) and placed in the white Led reactor (~23 W). When the reaction was complete, according to TLC or $^1$H NMR, the crude was poured into a funnel containing ethyl acetate (5 mL) and $H_2O$ (5 mL). The layers were separated; the organic phase was extracted with ethyl acetate (3 × 5 mL), washed with brine, dried over MgSO$_4$, and concentrated.

## Data availability

Materials and methods, detailed optimization studies, experimental procedures, mechanistic studies, UV–vis spectra are available within the article and the Supplementary Information and Supplementary Data Files. The experimental data

that support the findings of this study are available in 4TU.ResearchData with the identifier https://doi.org/10.4121/13186760.

CCDC 2021709 contains the supplementary crystallographic data for this paper. These data can be obtained free of charge via www.ccdc.cam.ac.uk/data_request/cif, or by emailing data_request@ccdc.cam.ac.uk, or by contacting The Cambridge Crystallographic Data Centre, 12 Union Road, Cambridge CB2 1EZ, UK.

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

## Acknowledgements

P.R. received financial support from EU through the MSCA Individual Fellowship program (MOSPhotocat, grant No. 793677). T.N. would like to acknowledge the Dutch Science Foundation (NWO) for a VIDI grant (SensPhotoFlow, grant No. 14150). M.P., M.F., and P.L. received the financial support from MINECO/FEDER (grants CTQ2015-69136-R and PID2019-109236RB-I00) and the CERCA Program/Generalitat de Catalunya. ICIQ thanks the Ministerio de Ciencia, Innovación y Universidades (Spain) for support through Grant PID2019-109236RB-I00 and Severo Ochoa Excellence Accreditation 2020-2023 (CEX2019-000925-S, MIC/AEI).

## Author contributions

P.R., T.N., and M.A.P. directed the project. P.R. designed and performed all the experiments and analyzed the data. M.F. performed all the computational studies. P.L. helped in carrying out and analyzing spectroscopic data. P.R., T.N., and M.F. wrote the paper with contributions from all authors.

## Competing interests

The authors declare no competing interests.
