## [Peer Review File · Nature Communications]

REVIEWER COMMENTS

Reviewer #1 (Remarks to the Author):

In this submission the authors study the Bi₂O₃ based catalyst and use a combination of experiment and computational methods to show that the actual catalyst is a BiBr species and the Bi₂O₃ is actually a pre-catalyst.

The identification of the active catalyst in a chemical reaction is a topical problem and one that still requires much work to be undertaken. This system is an interesting one to investigate this phenomenon and the authors present a comprehensive study of the problem.

For me this paper can be published in Nature Communications as the idea is of significant interest and the model system is well chosen.

I have some comments on the submission

1. Some minor details:

Introduction, a short explanation of photocatalysis would help

Introduction, define ATRA

Introduction, are Ru and Ir toxic?

Page 5, text "...also favour its coordination..." this is not easy to follow

Page 7, define "SMD"

2. In the abstract authors should mention that they use a brominated species and this is the origin of the BiBr

3. The title could lead a reader to think of heterogeneous photocatalysis (Similar to TiO₂ photocatalysis), but that is not really the case here.

4. In the computational section, authors do not state if an effective core potential was used for Bi and if so how many valence electrons are used. Given Bi is a heavy element do the potentials and/or basis sets account for relativistic effects?

5. Are there tabulated bond dissociation energies for C-halogens? (middle of page 5)

6. The figures for the modelling have no labels on the Y-axis.

7. I ask authors to review the modelling section and make it more clear to the reader how their results in the corresponding figures show the formation of Bi-Br containing species are more feasible. I am not 100% clear on what is presented in Figures 5 and 7 and authors could revise the presentation of these figures

Reviewer #2 (Remarks to the Author):

This paper describes study of true photocatalyst in ATRA reaction using Bi₂O₃ semiconductors by combination of theoretical and experimental studies.

The authors have searched and identified reactive species, homogeneous

Bi_nBr_m, in the ATRA reaction using

Bi₂O₃, and have elucidated the detailed mechanism of the photo heterogeneous reaction with many unclear points.

This paper contains very interesting content that overturns the previously ambiguous reaction mechanism. This reviewer considers this result to be very interesting and will be helpful when considering various reaction mechanisms in the future because it has a large spillover effect on other catalytic reactions.

On the other hand, this phenomenon can be considered to be specialized for this ATRA reaction,

and can be regarded as a very specific phenomenon.

Given versatility and the criteria of *Nature Communication*, this reviewer thinks that this phenomenon should be observed in at least one other reaction as well. Therefore, it cannot be accepted at this stage, but I recommend resubmitting to this journal after reconsidering it.

Reviewer #3 (Remarks to the Author):

This paper deals with one of organic chemical transformations by a heterogeneous photocatalyst Bi₂O₃ and proposes a catalytically active species in the atomic transfer radical addition (ATRA) reaction. The authors found that Bi₂O₃, insoluble in organic solvents, was dissolved during the ATRA reaction and claimed that the obtained soluble bismuth species should act as the active photocatalyst in this photocatalytic reaction and that Bi₂O₃ may play a role as a pre-catalyst. This is an interesting hypothesis, but the paper lacks experimental data to support it, and it seems premature to be published in *Nature Communications* at this stage.

The reviewer would like to propose an analysis by liquid chromatography (LC) for following the reaction kinetics, in which Bi₂O₃ changes in a soluble bismuth species during the reaction. LC using a photodiode array as a detector will be able to identify and separate species with characteristic absorption bands in the UV region, and by combining LC / MS, the species will be analyzed by molecular weight and fragment analysis. It may be possible to identify the species. How about proceeding with research by combining experimental data from LC and computational chemistry methods in this way?

Since the ATRA reaction occurs even if the solvent is changed, it seems necessary to explain the reaction mechanism consistent with this experimental fact. In this paper, X-ray structural analysis was carried out for the single crystal constituted of bismuth bromide with [(CH₃)₃S]⁺ cation derived from DMSO. If the authors claimed this bismuth halide species as the main active photocatalyst, how can the formation of bismuth species be achieved in other solvents (e.g., in acetonitrile)?

There are careless mistakes in this paper. For example, the symbols a – g in Fig. 4 do not match the description in the text.

REVIEWER COMMENTS and REBUTTAL

Reviewer #1 (Remarks to the Author):

In this submission, the authors study the Bi₂O₃ based catalyst and use a combination of experiment and computational methods to show that the actual catalyst is a BiBr species and the Bi₂O₃ is actually a pre-catalyst. The identification of the active catalyst in a chemical reaction is a topical problem and one that still requires much work to be undertaken. This system is an interesting one to investigate this phenomenon and the authors present a comprehensive study of the problem. For me this paper can be published in Nature Communications as the idea is of significant interest and the model system is well chosen. I have some comments on the submission.

- **Reviewer 1 comment:** Introduction, a short explanation of photocatalysis would help

– **Answer:** A brief definition of photocatalysis was already included in the previous version (with the corresponding references): *“In the last decade, we are appreciating a fast growth in the use of photocatalysis for the synthesis of chemical compounds. Along with it, a large variety of photocatalysts have been utilized which can convert sunlight into chemical energy and transfer this energy to the reacting molecules, thus effectively enabling light-fueled transformations.”* (ref. 1 and 2 - Organocatalysis Combined with Photocatalysis. *Top. Curr. Chem.* **377**:37, (2019), Illuminating Photoredox Catalysis. *Trends in Chemistry* **1**, 111-125, (2019)).

However, to accommodate the feedback of the referee, the following sentence was included in the introduction: *“Photocatalysis has emerged as a benchmark tool that combines light (in an appropriate wavelength) and a (photo-)catalyst to carry out chemical transformations that are otherwise impossible using standard synthetic procedures.”*

- **Reviewer 1 comment:** Introduction, define ATRA

– **Answer:** The sentence “...atom transfer radical addition (ATRA)-type reactions.” was added in the introduction

- **Reviewer 1 comment:** Introduction, are Ru and Ir toxic?

– **Answer:** Some compounds/complexes of ruthenium are known to be hazardous for human and aquatic environments (for more information, see: Chem. -Bio. Interactions Mutagenic and toxic effects of ruthenium, 1980, 31, 355). However, little is known about the toxicity of iridium metal and iridium complexes, which are used in photocatalysis. However, it is fair to say that these complexes might have

some negative biological impact in the human body. Nevertheless, to avoid having to give a specific statement about the toxicity of these metals, we decided to remove the word toxicity from the main manuscript. So the sentence: “*Moreover, in some cases, it can replace the use of expensive and toxic Ru- and Ir- based transition metal complex photocatalysts*” was replaced by “*Moreover, in some cases, it can replace the use of metal complexes based on expensive and non-abundant Ru and Ir transition metal photocatalysts.*”

- **Reviewer 1 comment:** Page 5, text ".also favor its coordination..." this is not easy to follow
 - **Answer:** The sentence on page 5: “*However, the soft Lewis acid character and the halogenophilic properties of bismuth, also favor its coordination with an ionizable halide atom acting as a bromine ion acceptor.*” was replaced by: “*However, the soft Lewis acid character and the halogenophilic properties of bismuth, may result in the coordination with the bromine atom.*”

- **Reviewer 1 comment:** Page 7, define "SMD"
 - **Answer:** The SMD abbreviation was defined in the main text: “Solvation Model Based on Density (SMD)”. This solvation scheme belongs to the family of IEF-PCM solvations developed by Truhlar and co-workers (see refs. 5, 6, 9 Supplementary: Truhlar et al. *J. Chem. Phys.* **125**, 194101, (2006); *Chem. Phys. Lett.* **502**, 1-13, (2011); *J. Phys. Chem. B*, **113**, 6378-6396 (2009)). It is generally considered very accurate in calculating free energies of solvation.

- **Reviewer 1 comment:** In the abstract authors should mention that they use a brominated species and this is the origin of the BiBr
 - **Answer:** The sentence; “*Herein, we show through a combination of theoretical and experimental studies that the perceived heterogeneous photocatalysis with Bi₂O₃ actually involves a homogeneous Bi_nBr_m species’* was replaced by “*Herein, we show through a combination of theoretical and experimental studies that the perceived heterogeneous photocatalysis with Bi₂O₃ and in the presence of alkyl bromides involves a homogeneous Bi_nBr_m species’.*

- **Reviewer 1 comment:** The title could lead a reader to think of heterogeneous photocatalysis (Similar to TiO₂ photocatalysis), but that is not really the case here.
 - **Answer:** We appreciate the comment, however after a detailed discussion about the title with all co-authors, we have decided to keep it as it is. The reason is, that in our opinion, the title reflects

perfectly the content of our manuscript. The word heterogeneous is not even mentioned and thus we do not see the issue.

- **Reviewer 1 comment:** In the computational section, authors do not state if an effective core potential was used for Bi and if so how many valence electrons are used. Given Bi is a heavy element do the potentials and/or basis sets account for relativistic effects?

– **Answer:** The referee is right, we did not state clearly the use of ECP in the SI, but we did indeed use the Weigend-Ahlrichs' ECP. Def2-SVP and Def2-TZVPP (and the congeners) are full electron basis sets until the end of the 4th period (Kr), then they rely on small-core ECP to “freeze” out the core electrons and to introduce scalar relativistic effects (see ref. 8. Supplementary: *Phys. Chem. Chem. Phys.* **7**, 3297-3305, (2005)). Please, find the specs for the ECP used with Bi right below. In the case of Bismuth 60 electrons (till 4f) have been modeled by ECP and 23 ($6d^2 + 6p^3 + 5d^{10}$) valence + sub-valence ($5s^2 5p^6$) electrons have been modeled by the basis.

```
=====
=
                Pseudopotential Parameters
=====
=
Center   Atomic   Valence   Angular   Power
Number   Number   Electrons Momentum of R   Exponent   Coefficient SO-Coefficient
=====
=
  1      83      23
      H and up
      2      1.0000000    0.00000000  0.00000000
      S - H
      2      13.0430900    283.26422700  0.00000000
      2      8.2216820     62.47195900  0.00000000
      P - H
      2      10.4677770    72.00149900 -144.00299800
      2      9.1189010     144.00227700  144.00227700
      2      6.7547910     5.00794500 -10.01589000
      2      6.2525920     9.99155000  9.99155000
      D - H
      2      8.0814740     36.39625900 -36.39625900
      2      7.8905950     54.59766400  36.39844300
      2      4.9555560     9.98429400 -9.98429400
      2      4.7045590     14.98148500  9.98765700
      F - H
      2      4.2145460     13.71338300 -9.14225600
      2      4.1334000     18.19430800  9.09715400
      G - H
      2      6.2057090    -10.24744300  5.12372200
      2      6.2277820    -12.95571000 -5.18228400
```

The choice of this ECP and basis set(s) was motivated by the need to have an overall basis set able to describe both the “inorganic” and the “organic” part of the complexes. This ECP/basis combo has been already used in the chemistry of bismuth recently (Schulz et al. *Inorg. Chem.* 59, 3344-3352, (2020); *Dalton Trans.* 44, 14589-14593, (2015); *Dalton Trans.* 49, 11756-1177, (2020).

Please notice that the information concerning the ECP has been updated in the revised Supplementary of the manuscript at the beginning of the computational section.

- **Reviewer 1 comment:** Are there tabulated bond dissociation energies for C-halogens? (middle of page 5)

- **Answer:** The bond dissociation energies reported were obtained from reference 33 as already cited in the main manuscript (ref. 33 - Darwent, B. deB. Bond Dissociation Energies in Simple Molecules; National Bureau of Standards: Washington, DC, 1970).

- **Reviewer 1 comment:** The figures for the modelling have no labels on the Y-axis.

- **Answer:** Figures 5 and 7 have been redone (according to the next comment) and the labels have been added.

- **Reviewer 1 comment:** I ask authors to review the modelling section and make it more clear to the reader how their results in the corresponding figures show the formation of Bi-Br containing species are more feasible. I am not 100% clear on what is presented in Figures 5 and 7 and authors could revise the presentation of these figures.

- **Answer:** Figures 5 and 7 have been redone following the suggestions of the referee. The content in Figure 5 has been reduced and only 8 out of 48 entries have been incorporated. We chose these entries because we think they are the most relevant within our data pool to explain methodologies and results. We believe both Figure 5 and 7 are very clear at glance now and stave off confusion. Concomitantly, we adjusted the computational section, from line #3 of page #8 to line #8 of page #9 and from line #15 of page #9 to the end of the section.

The text was changed according to the new figures and the explanation has been enhanced and further clarified.

Reviewer #2 (Remarks to the Author):

This paper describes the study of the true photocatalyst in ATRA reaction using Bi_2O_3 semiconductors by a combination of theoretical and experimental studies. The authors have searched and identified reactive species, homogeneous BinBr_m, in the ATRA reaction using Bi_2O_3 , and have elucidated the detailed mechanism of the photo heterogeneous reaction with many unclear points. This paper contains very interesting content that overturns the previously ambiguous reaction mechanism. This reviewer considers this result to be very interesting and will be helpful when considering various reaction mechanisms in the future because it has a large spillover effect on other catalytic reactions. On the other hand, this phenomenon can be considered to be specialized for this ATRA reaction and can be regarded as a very specific phenomenon. Given the versatility and the criteria of Nature Communication, this reviewer thinks that this phenomenon should be observed in at least one other reaction as well. Therefore, it cannot be accepted at this stage, but I recommend resubmitting this journal after reconsidering it.

– **Answer:** We agree with the referee that, to reach the level of Nature Communications, the versatility of the method/study has to be demonstrated. The formation of the homogeneous species takes place as a result of the reaction between electron-deficient alkyl bromides and Bi_2O_3 in different solvents (as demonstrated in the research work). Thus, it is expected that in any reaction which involves these two compounds a homogeneous species of BinBr_m (solv.) is involved. The formation of the homogeneous species was also observed in other transformations such as in the α -alkylation of aldehydes as already cited in the main manuscript: *“Importantly, in both the α -alkylation of aldehydes and the Kharasch addition, a phase change can be observed in the reaction mixture while the reaction progresses (Fig.1). The reaction mixture shifts from a suspension to a transparent yellowish solution.”* The homogenization of the Bi_2O_3 was already briefly reported in a previous publication (ref. 14: Light-Driven Organocatalysis Using Inexpensive, Nontoxic Bi_2O_3 as the Photocatalyst. *Angew. Chem. Int. Ed.* **53**, 9613-9616, (2014)). We detected this species by UV-Vis in both reactions, but finally, we decided to use the ATRA reaction as a model due to its lower complexity compared to the α -alkylation (bases are required). Besides given the communication character of the journal, the homogenization of the Bi_2O_3 in the α -alkylation will be soon reported with more details in another publication. Based on that, we believe that the study of the photocatalytic species using as a model the ATRA reaction demonstrates the behavior of the Bi_2O_3 in the presence of electron-deficient alkyl halides and it could be extended to other systems where these compounds are present (Bi_2O_3 /alkyl halides/solvents).

Reviewer #3 (Remarks to the Author):

This paper deals with one of organic chemical transformations by a heterogeneous photocatalyst Bi₂O₃ and proposes a catalytically active species in the atomic transfer radical addition (ATRA) reaction. The authors found that Bi₂O₃, insoluble in organic solvents, was dissolved during the ATRA reaction and claimed that the obtained soluble bismuth species should act as the active photocatalyst in this photocatalytic reaction and that Bi₂O₃ may play a role as a pre-catalyst. This is an interesting hypothesis, but the paper lacks experimental data to support it, and it seems premature to be published in Nature Communications at this stage. The reviewer would like to propose an analysis by liquid chromatography (LC) for following the reaction kinetics, in which Bi₂O₃ changes in a soluble bismuth species during the reaction. LC using a photodiode array as a detector will be able to identify and separate species with characteristic absorption bands in the UV region, and by combining LC / MS, the species will be analyzed by molecular weight and fragment analysis. It may be possible to identify the species. How about proceeding with research by combining experimental data from LC and computational chemistry methods in this way?

– **Answer:** Although we do not disagree with the referee that the use of LC/MS might potentially give valuable information about the formation of a homogeneous species, we have not done this experiment for the following reasons:

- 1) Since bismuth-based compounds have a large atomic weight (Bi: 208.98), we have serious doubts if the LC/MS could detect these species.
- 2) We do not have access to this type of equipment in our research labs;
- 3) Finally and most importantly, in our opinion, the experiments carried out in our research (supported by the theoretical study) lead to solid results that are enough to sustain our study.

• **Reviewer 3 comments:** Since the ATRA reaction occurs even if the solvent is changed, it seems necessary to explain the reaction mechanism consistent with this experimental fact.

– **Answer:** We concluded that BiBr₃ is a plausible catalytic species for the ATRA reaction in the computational study presented in Figure 5. BiBr₃ cannot exist (as it is) in condensed phase for two main reasons: 1) coordinative unsaturation; 2) the presence of strong σ -donor solvents, like acetonitrile or DMF or DMSO. The σ -donation from solvent moieties is necessary to complete the Bi coordination sphere, thus stabilizing BiBr₃ and prompting the formation of solvato-complexes. We calculated possible solvato-structures in presence of DMSO in Figure 7. The referee may notice that bismuth solvato-complexes of equal structures (monomers and dimers) and different halogen have been already reported crystallographically, due to their stability (ref. 15 Supplementary - *Austr. J. Chem.* **51**, 285-292, (1998)). The referee may notice that our calculations, generated independently and aprioristically

from the above reference, are nonetheless convergent with it and lead to the same structural possibilities. The referee may also notice that we did not suggest a single solvato-structure, yet we proposed a group of them since they are all competitively forming and decaying based on thermodynamic equilibria in solution.

This said, it is highly probable that BiBr₃ forms the same solvato-structures in presence of acetonitrile or DMF as well; after all DMF can act as a σ,σ -bidentate donor ligand, forming monomeric or dimeric complexes (just like DMSO) and acetonitrile can act as a σ -donor and π -donor ligand, forming again monomers or dimers. Even though they have not been characterized crystallographically (in our opinion due to inferior stability compared to the DMSO solvato-complexes), species like Br₃Bi(NCMe)₃ or Br₃Bi(DMF)₃ are very plausible to exist in solution. The referee should also notice the reference (Solvation of the Bismuth(III) Ion by Water, Dimethyl Sulfoxide, N,N'-Dimethylpropyleneurea, and N, N-Dimethylthioformamide. An EXAFS, Large-Angle X-ray Scattering, and Crystallographic Structural Study. *Inorg. Chem.* **39**, 4012-4021, (2000)), describing the full X-ray characterization of Bi complexes with DMSO and DMF-like ligands and showing very similar structures to those proposed in the present manuscript. Besides, the formation of the Bi-based homogeneous species was experimentally detected by UV-Vis spectroscopy in different solvents, as cited in the main manuscript: “*A similar interaction with α -bromo esters was also observed in the presence of other solvents (Supplementary Fig. S4).*” As expected, UV-Vis peaks change in intensity, shift, and patterns in different solvents; these are clear signs that the vertical excitations are affected by the different solvation of the primary coordination sphere of Bi³⁺ in BiBr₃. We modified the main manuscript from line #15 of page #9 to the end of the section to include plausible speculations in the formation of solvated BiBr₃ complexes in other solvents different from DMSO. The special case represented by the simultaneous presence of Bi₂O₃/DMSO/DEBM in the reaction vessel is discussed in the next comment.

- **Reviewer 3 comments:** In this paper, X-ray structural analysis was carried out for the single crystal constituted of bismuth bromide with [(CH₃)₃S]⁺ cation derived from DMSO. If the authors claimed this bismuth halide species as the main active photocatalyst, how can the formation of bismuth species be achieved in other solvents (e.g. in acetonitrile)?

– **Answer:** This is a very good question and we thank the referee for giving us the chance to further demonstrate our reasoning based on experimental/theoretical evidence. We cleared the role of stabilization lead by σ -donor solvents on BiBr₃ in the previous comment. In summary, any solvent able to medium-to-strong σ -donation can solvate the unit “BiBr₃” giving monomers and dimers in solution. When the solubilization of Bi₂O₃ takes place in the simultaneous presence of DMSO and DEBM, however, we are not able to isolate and crystallize any solvato-complex as in ref. 15 supplementary (Syntheses, Structures and Vibrational Spectra of Some Dimethyl Sulfoxide Solvates of Bismuth(III))

Bromide and Iodide. *Austr. J. Chem.* **51**, 285-292, (1998)). The theoretical model provides an excellent explanation (highlighted in Figure 7). $[(\text{CH}_3)_3\text{S}]^+\text{Br}$, the degradation product of DMSO in presence of DEBM, acts as a ligand itself for the BiBr_3 (Br^- coordinates Bi for its high affinity and the SMe_3^+ acts as a counter-ion), displacing any possible coordinated molecules of DMSO. The resulting compound, $[(\text{CH}_3)_3\text{S}]_3[\text{BiBr}_6]$, is a **thermodynamic product** due to its excessive stability compared to solvato-complexes. Theory predicts marvelously why we should be able to isolate and even crystallize **only $[(\text{CH}_3)_3\text{S}]_3[\text{BiBr}_6]$** (and not other species).

This is, however, a very special case arising from the simultaneous presence of DMSO and DEBM. **In other solvents, the ATRA reaction can be carried out by monomeric or dimeric solvato-complexes of BiBr_3 .**

We chose DMSO for this present work because it was the chosen solvent for the previous ATRA reaction, already reported by our group (ref. 17 - Visible Light-Driven Atom Transfer Radical Addition to Olefins using Bi_2O_3 as Photocatalyst. *ChemSusChem* **8**, 1841-1844, (2015)), and we thought that this fact would enable a straight and convenient comparison with the previous paper.

We introduced modifications to the main article from line #15 of page #9 to the end of the section to clarify the doubts raised by the referee and introduce rational speculation of the solvation in other solvents.

- **Reviewer 3 comments:** There are careless mistakes in this paper. For example, the symbols a – g in Fig. 4 do not match the description in the text.

- **Answer:** The mistakes were fixed in the main manuscript.

REVIEWER COMMENTS

Reviewer #1 (Remarks to the Author):

The authors have addressed the comments I raised for the first version of this submission. For this reviewer, the submission is now acceptable for publication, if the other reviewers are also satisfied with the changes.

Reviewer #3 (Remarks to the Author):

The revised manuscript is well amended. The present reviewer agrees with the authors' claim regarding the presence of the Bi species in DMSO. However, even if the existence of the solvated Bi species in other solvents (DMF, DMA, or acetonitrile) is certainly suggested by the cited paper e.g. *Inorg. Chem.* 39, 4012, 2000, it seems to be still unclear. The presence of Bi species derived from BiBr₃ promotes the ATRA reaction in DMSO as shown in Fig. 4g and Fig. 4h. It is presented as evidence of involvement of the Bi species in this type of reaction. If so, similar studies should be made in other solvents. The present reviewer thinks that the revised manuscript should be published in *Nature Communications* after addressing or considering the above point.

Reviewer #3 (Remarks to the Author):

The revised manuscript is well amended. The present reviewer agrees with the authors' claim regarding the presence of the Bi species in DMSO. However, even if the existence of the solvated Bi species in other solvents (DMF, DMA, or acetonitrile) is certainly suggested by the cited paper e.g. Inorg. Chem. 39, 4012, 2000, it seems to be still unclear. The presence of Bi species derived from BiBr₃ promotes the ATRA reaction in DMSO as shown in Fig. 4g and Fig. 4h. It is presented as evidence of involvement of the Bi species in this type of reaction. If so, similar studies should be made in other solvents. The present reviewer thinks that the revised manuscript should be published in Nature Communications after addressing or considering the above point.

Answer: Actually, we have this information already in our manuscript. The UV-vis spectra of the ATRA reaction already demonstrates the presence of this solvated Bi species in other solvents, such as DMF, DMI, DMA, and CH₃CN (Supplementary Fig. S4b,c). To complement this study, we have included in the Supplementary two more experiments:

1. In the first experiment (Supplementary Fig. S4d), we show the UV-vis of the ATRA reaction photocatalyzed by Bi₂O₃ and BiBr₃ in DMF. The data was plotted together with the UV-vis spectrum of BiBr₃ in DMF. In all cases, absorption bands with very similar patterns were detected near the visible region by UV-vis spectroscopy confirming the formation of a Bi³⁺ solvated species in DMF.

2. In the second experiment (Supplementary Fig. S4e), we show a kinetic study of the ATRA reaction photocatalyzed by Bi₂O₃ in DMF. The results were compared to the same reaction in DMSO. In both cases, it is observed an S-shape tendency of the curve suggesting a preliminary induction period that corroborates the formation of a (photocatalytic) soluble species.

Also, a new sentence was added to the main manuscript, referring to the new experiments included in the Supplementary: "A similar behavior was observed when the reaction was carried out in presence of DMF (Supplementary Fig, S4e)."

We believe that the new results (together with previous ones, Supplementary Fig. S4b, and c) demonstrate that the formation of this species is also possible in the presence of other solvents.